# The utility of *Escherichia coli* as a contamination indicator for rural drinking water: Evidence from whole genome sequencing

**Saskia Nowicki**[1]*, **Zaydah R. deLaurent**[2], **Etienne P. de Villiers**[2,3,4], **George Githinji**[2], **Katrina J. Charles**[1]

1 School of Geography and the Environment, University of Oxford, Oxford, United Kingdom, 2 KEMRI-Wellcome Trust Research Programme, Centre for Geographic Medicine Research-Coast, Kilifi, Kenya, 3 Nuffield Department of Medicine, Centre for Tropical Medicine and Global Health, University of Oxford, Oxford, United Kingdom, 4 Department of Public Health, Pwani University, Kilifi, Kenya

* saskia.nowicki@ouce.ox.ac.uk

**Data Availability Statement:** The raw sequence read files have been deposited in the European Nucleotide Archive (ENA) as study accession

## Abstract

Across the water sector, *Escherichia coli* is the preferred microbial water quality indicator and current guidance upholds that it indicates recent faecal contamination. This has been challenged, however, by research demonstrating growth of *E. coli* in the environment. In this study, we used whole genome sequencing to investigate the links between *E. coli* and recent faecal contamination in drinking water. We sequenced 103 *E. coli* isolates sampled from 9 water supplies in rural Kitui County, Kenya, including points of collection (n = 14) and use (n = 30). Biomarkers for definitive source tracking remain elusive, so we analysed the phylogenetic grouping, multi-locus sequence types (MLSTs), allelic diversity, and virulence and antimicrobial resistance (AMR) genes of the isolates for insight into their likely source. Phylogroup B1, which is generally better adapted to water environments, is dominant in our samples (n = 69) and allelic diversity differences (z = 2.12, p = 0.03) suggest that naturalised populations may be particularly relevant at collection points with lower *E. coli* concentrations (<50 / 100mL). The strains that are more likely to have originated from human and/or recent faecal contamination (n = 50), were found at poorly protected collection points (4 sites) or at points of use (12 sites). We discuss the difficulty of interpreting health risk from *E. coli* grab samples, especially at household level, and our findings support the use of *E. coli* risk categories and encourage monitoring that accounts for sanitary conditions and temporal variability.

## Introduction

The gram-negative coliform bacterial species, *Escherichia coli*, was discovered in 1885 during an investigation of the microbial life of the human gastrointestinal tract [1]. On the basis of their association with the gastrointestinal tract, *E. coli* have been recommended as indicators of microbial contamination risk in water since 1892 [2]. For decades, they were studied

PRJEB40218 (http://www.ebi.ac.uk/ena/data/view/PRJEB40218).

**Funding:** S.N. and K.C. are supported by the REACH programme funded by UK Aid from the UK Foreign, Commonwealth and Development Office (FCDO | https://www.gov.uk/government/organisations/foreign-commonwealth-development-office) for the benefit of developing countries (Programme Code 201880). S.N. is further supported by the Commonwealth Scholarship Commission in the UK (https://www.gov.uk/government/organisations/commonwealth-scholarship-commission-in-the-uk). G.G. is funded by the National Institute for Health Research (project reference 17/63/82) funded by UK Aid (FCDO) to support global health research. However, the views expressed and information contained in this article are not necessarily those of or endorsed by the FCDO, which can accept no responsibility for such views or information, or for any reliance placed on them. The funders had no role in study design, data collection and analysis, decision to publish, or preparation of the manuscript.

**Competing interests:** The authors have declared that no competing interests exist.

primarily in lab and clinical settings that further emphasised their association with the gastro-intestinal tract and, therefore, the faeces of humans and other vertebrates. But their ecological niche is much broader than initially realised, and the species is now recognised as 'hardy generalist' rather than gastrointestinal tract specialist [3]. This has consequences for its use as a water quality indicator.

*E. coli* occupy two habitats: the primary [gastrointestinal tract) and secondary [water, sediment, soil, flora). Initially, they were thought to have a net negative growth rate in the secondary habitat, implying short-term extra-host persistence [4]. But since the early 2000s, the assumption of negative growth rate in secondary habitats has increasingly been challenged by recognition of naturalised *E. coli* populations. *E. coli* have a core genome of about 2,000 genes, but roughly half the genes in any *E. coli* bacterium are contained in the 'accessory genome'–which exhibits large genotypic and phenotypic diversity across strains, allowing for diverse adaptive paths [5, 6]. This adaptability enables *E. coli* to contend with many stresses in environmental habitats [7, 8], which could include nutrient deprivation; sub-optimal temperature, salinity, moisture, and substrate texture ranges; exposure to solar radiation; competition with other microbes; and predation by protozoans [9, 10]. *E. coli* have been found to grow in solutions of pH ranging from 4.5 to 9 [11]. They are able to acquire energy in versatile ways and grow in both anaerobic and aerobic conditions with temperatures ranging from 7.5 to 49˚C [10].

Studies have demonstrated *E. coli* persisting on algae and in soils, sediment, and sand in tropical, subtropical and temperate environments [10], as well as in handpumps removed from boreholes [12]. Furthermore, long-term growth of *E. coli* has been demonstrated in a diverse array of source environments including lakes, rivers, sediments, beaches, soils, and aquatic plants and animals [13]. Several investigations have described genetically distinct populations of naturalised *E. coli* [14–20], including environmentally associated cryptic clade phylogroups [13]. And a recent meta-analysis representing the phylogenetic diversity of more than 5000 (mostly) non-clinical isolates from Australia associated genetic backgrounds with specific habitats, including freshwater [21].

Such a meta-analysis is not yet possible for Kenya, nor the African continent more broadly, because genetic studies of non-clinical *E. coli* are relatively sparse. The majority of *E. coli* isolates sequenced in Kenya are from clinical, livestock, or food hygiene sampling [22], so there is limited information on the genetic background of *E. coli* sampled from the environment, including water systems. In general, few studies have undertaken whole genome sequencing of *E. coli* in drinking water supplies, although a characterisation of 28 isolates from chlorinated water supplies in Australia suggested that 9 isolates were naturalised *E. coli* that were unlikely to represent human health risk [23].

The utility of *E. coli* as a faecal contamination indicator in water systems is challenged by the existence of naturalised *E. coli* populations [24]. It can be difficult to interpret *E. coli* sampling results, particularly when water system safety controls have not been validated and when sampling is infrequent [25]. Thus, the severity and immediacy of the potential threat indicated by an *E. coli* positive sample is unclear. Despite this uncertainty, *E. coli* remains the preferred indicator of microbial water safety, including for rural community managed supplies [26]. The World Health Organisation drinking-water quality guidelines do not account for naturalised *E. coli* populations in source waters or distribution systems; they continue to assure that "the presence of *E. coli* should be considered as evidence of recent faecal contamination" [26 p57].

In tracking progress towards the drinking-water Sustainable Development Goal (SDG 6.1), *E. coli* is used to assess whether water is free from faecal contamination, a prerequisite of being considered 'safely managed'. As a result, *E. coli* testing is included in large-scale household survey programmes in more than thirty countries [27]. These survey programmes are testing

water both at point of collection (PoC) and household-level point of use (PoU). That concentrations of *E. coli* often increase between PoC and PoU is well-established in the literature, evinced by studies from both urban and rural areas in Africa, South-East Asia, South and Central America, and the Caribbean [28–30]. However, the health implications of this increase are debated, as are the implications for where water quality monitoring and treatment efforts should focus [30]. Increases in *E. coli* concentration between PoC and PoU may result from contamination introduced during transport and household storage, which is influenced by sanitation, hygiene, and water handling practices; however, it may also result from growth of *E. coli* that was in the water supply, or from sloughing of biofilms in storage containers–neither of which would normally represent an increased health hazard [31, 32].

Thus, use of *E. coli* in rural water quality monitoring involves two key uncertainties. First, to what extent is *E. coli* at PoC linked to recent faecal contamination? Second, what are the health hazard implications of changes in *E. coli* concentration between PoC and PoU? These long-standing [33] and strongly context dependant uncertainties are not easily resolved. In this study, we sought insight for managing them in the context of water supply in rural Kitui County, Kenya. We isolated *E. coli* from drinking water supplies and household storage and used whole genome sequencing to characterise each isolate to investigate the diversity of phylogroups, multi-locus sequence types, and virulence and antibiotic resistance genes within and between water supplies and household water storage. The results contribute to understanding of non-clinical *E. coli* populations in Kenya and provide insight into the occurrence of *E. coli* in rural water systems, allowing comparison between PoC and PoU and informing on the utility of *E. coli* as a faecal contamination indictor in these settings.

## Materials and methods

### Ethics statement

Field research was conducted with permission from the Kenyan National Commission for Science, Technology and Innovation, License No: NACOSTI/P/18/22793/24854. Ethical approval was obtained from the University of Oxford's Central University Research Ethics Committee. All participation was informed and uncompensated. Engagement with participants was contingent upon consent from participants after they were informed of the study process and objective verbally. Personal identifiers were stored only for the duration of the study and in a secured platform.

### Water sampling

Nine water supply systems were selected from among those that were sampled by a rural water maintenance service provider in northern Kitui County, Kenya as part of a monitoring programme that had commenced 7 months prior to this study. The systems were chosen to include multiple supply types with varying degrees of protection against contamination (piped schemes sourcing water from boreholes or reservoirs and point sources including boreholes and hand-dug wells with or without handpumps). To be selected, the system had to be a main or alternative source of drinking-water with above zero median *E. coli* concentration from the monitoring programme results. Most of the selected systems are community managed and are either point sources or small (<5 km) piped distribution systems, all of which include one or more storage tanks. Only one of them (S5) is managed by a formal water service provider (utility). The water in this system is clarified and chlorinated in a water treatment facility and then piped through a 66 km distribution network. The utility produces about 3000 m$^3$/day on average, but due to the size of the distribution system and high demand for water, the supply at the PoC is intermittent (approximately 2 days / week).

For each system, we sampled one or more PoCs. For each PoC, we sampled multiple PoUs. We asked water system managers and users to help us find homes with stored water. The goal was to sample at least three PoUs per PoC, but it was sometimes difficult to find households with stored water from the PoC of interest due to multiple source use and mixing water from multiple sources in home storage. As indicated in Table 1, a total of 44 sites were sampled including 14 PoCs and 30 PoUs (more details on site protection and storage length and location are available for each site in S1 Table). We refer to a system and its associated PoCs and PoUs as a set, and each sampling site is labelled according to its corresponding set, PoC and PoU (S#C#U#). The sampling sites are located within 1,400 km$^2$, with the distance between any two PoCs from different systems ranging from 50 km to 15 m apart.

Seven months of monitoring data were available for the selected PoC sites, which had been sampled either weekly or monthly (with some gaps due to breakdowns and dry periods). Monitoring included *in situ* measurement of pH and conductivity using a HACH multimeter (HQ 40D), which was calibrated weekly, and *E. coli* sampling using an IDEXX Quanti-Tray2000 system with weekly field blank and duplicate samples. No *E. coli* was detected in the blank samples and duplicates had an average relative percent difference of 25.6% with 88% of duplicate pairs indicating the same risk category, as defined by the World Health Organisation [26]. Surveys and interviews were conducted alongside the monitoring activities to understand water use practices and track managers' responses to the test results.

The median pH for each site ranged from 6.6 to 8.6 and median conductivity ranged from 90 to 1600 μS/cm except for Sets 4 and 7, which had median conductivity of 5000 and 10,500 μS/cm, respectively. Due to the high conductivity, which is linked to salinity, the managers reported that water from these sites is only used for drinking when better alternatives are unavailable due to dry periods, breakdowns, affordability, and intermittent supply from the utility. Box-and-whisker plots for each PoC are available for pH (S1 Fig) and conductivity (S2 Fig). The median *E. coli* concentrations ranged from 1 to 920 MPN/100mL, with all PoCs having variable results over the monitoring period (Fig 1).

No previous sampling was available for the PoU sites. During the sampling visits for this study, short surveys were conducted with the household water managers (as distinct from household heads who are not always aware of the details of water management in the home). Respondents reported that PoU water was transported by donkey, motorbike, wheelbarrow, or self-carry to be stored in either plastic drums (approximately 200L) or jerricans (20L) located either inside the main house, a separate shelter, or outside in a private or communal area; at the time of sampling, the water was reported to have been stored between 0 and 4 days, with 1 or 2 days being most common (S1 Table). Respondents said that jerricans were occasionally cleaned with sand; they did not approximate a washing schedule but said the decision to clean depends on the colour inside the jerrican. Although the first household in set 8 reported occasional boiling and the third in set 6 reported occasional chlorination, none had treated the water that we sampled. PoU water was sampled in line with standard practice by asking the respondent to provide us with a cup of water that they would normally drink from.

Sites within a set were sampled on the same day between July 18 and August 2, 2019, which is dry season in Kitui. This season was chosen to control for rain events, which would differentially impact the quality of PoC water versus PoU water that was stored for multiple days. Two samples were collected at each site in sterile 100mL Whirl-pak bags with sodium thiosulphate to neutralise residual chlorine. At the PoCs, taps and handpump spouts were disinfected by flame and at least 20L of water was pumped or flushed prior to sample collection.

**Table 1. List of sampling sets with site IDs and source, system, collection, transport and storage types.**

| Set | Source and system | Collection type | Site ID | Transport type | Storage type |
|---|---|---|---|---|---|
| 1 | Reservoir piped | Standpipe | S1C1 | | |
| | | | S1C1U1 | Motorbike | Jerrican |
| | | | S1C1U2 | Donkey | Drum |
| | | | S1C1U3 | Donkey | Drum |
| | | | S1C1U4 | Donkey | Jerrican |
| | | | S1C1U5 | Donkey | Jerrican |
| | | | S1C1U6 | Donkey | Jerrican |
| | | Standpipe | S1C2 | | |
| | | | S1C2U1 | Donkey | Drum |
| | | | S1C2U2 | Donkey | Jerrican |
| | | | S1C2U3 | Donkey | Jerrican |
| 2 | Borehole piped | Standpipe | S2C1 | | |
| | | | S2C1U1 | Wheelbarrow | Drum |
| | | | S2C1U2 | Self-carry | Jerrican |
| | | Standpipe | S2C2 | | |
| | | | S2C2U1 | Motorbike | Jerrican |
| | | | S2C2U2 | Motorbike | Drum |
| | | | S2C2U3 | Wheelbarrow | Drum[a] |
| | | School standpipe | S2C3[b] | | |
| 3 | Borehole piped[2] | Standpipe | S3C1 | | |
| | | | S3C1U1 | Self-carry | Jerrican |
| | | | S3C1U2 | Self-carry | Drum |
| | | | S3C1U3 | Donkey | Drum |
| | | | S3C1U4 | Self-carry | Jerrican |
| | | School concrete tank | S3C2[b] | | |
| 4 | Borehole piped | Plastic tank | S4C1 | | |
| | | | S4C1U1 | Self-carry | Jerrican |
| | | | S4C1U2 | Wheelbarrow | Jerrican |
| | | Concrete tank | S4C2[c] | | |
| 5 | Reservoir piped[d] | Plastic tank | S5C1 | | |
| | | | S5C1U1 | Donkey | Jerrican |
| | | | S5C1U2 | Donkey | Drum |
| | | | S5C1U3 | Donkey | Jerrican |
| 6 | Borehole point | Handpump | S6C1 | | |
| | | | S6C1U1 | Donkey | Jerrican |
| | | | S6C1U2 | Donkey | Drum |
| | | | S6C1U3 | Donkey | Jerrican |
| 7 | Borehole point | Handpump | S7C1 | | |
| | | | S7C1U1 | Donkey | Jerrican |
| | | | S7C1U2 | Self-carry | Jerrican |
| 8 | Dug well point | Bucket draw | S8C1 | | |
| | | | S8C1U1 | Donkey | Jerrican |
| | | | S8C1U2 | Donkey | Jerrican |
| 9 | Borehole piped | School concrete tank | S9C1[b] | | |

[a]This storage drum had a dispensing tap.

[b]The PoCs in schools (n = 3) were not associated with PoU storage because students drank directly from the standpipes or tank taps.

[c]This PoC was sampled twice, two weeks apart, but no PoU water was available either time.

[d]Chlorine is used for treatment in this system, but dosing is inconsistent and chlorine residual testing is not done.

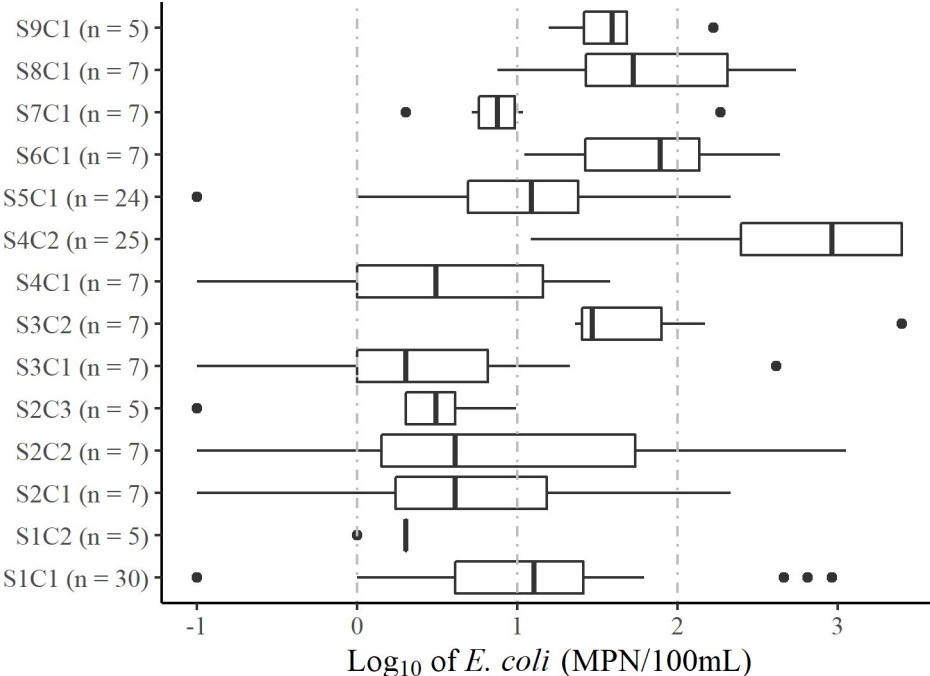

**Fig 1. Box-and-whisker plot of log$_{10}$-transformed monitoring programme *E. coli* results for selected PoCs.** The boxes show median values and span lower to upper quartiles, the whiskers show the lowest and highest datums within 1.5 times the interquartile range. Data are censored by a lower bound of 0 MPN/100 mL and an upper bound of > 2419.6 MPN/100mL. Results at the lower bound were converted to 0.1 (log = -1) and results at the upper bound were converted to 2500 (log = 3.4). Vertical dashed grey lines show the WHO recommended risk category cut-offs [26] with *E. coli* equal to 1, 10, and 100 MPN/100mL (corresponding to log values of 0, 1, and 2). The risk categories are low (<1), intermediate (1–9), high (10–99), and very high (>99).

## Sample filtration, culturing, and preservation

Samples were transported in a cooler box with icepacks to a field-lab where they underwent membrane filtration. Based on previous *E. coli* monitoring results, multiple dilutions were used for each site to maximise the chance of growing well-isolated colonies. Following filtration, the samples were incubated with m-ColiBlue24 broth (EPA Approved Hach Co.: 10029 method), which indicates *E. coli* colonies by blue colouration resulting from hydrolysis of 5-bromo-4-chloro-3-indolyl-β-D-glucuronide (BCIG). In all cases, the time between sample collection and filtration was less than 6 hours. Between filtration and incubation, samples had resuscitation time of 1 to 4 hours. Samples were incubated at 44.5˚C for 18–24 hours. The incubation temperature was chosen to discourage growth of non-thermotolerant coliforms and improve isolation of *E. coli* colonies. We considered that incubation at 44.5˚C could disadvantage naturalised *E. coli*, but studies have shown that although environmental strains grow better than enteric strains at low temperatures, their maximal growth rate and optimal temperature for growth are not distinct from enteric strains [34, 35].

Up to 6 colonies per site were selected for streaking on agar plates (ReadyPlate CHROM Chromocult Coliform Agar ISO 9308–1:2014), which inhibit growth of non-coliforms and distinguish *E. coli* based on β-glucuronidase activity. The plates were incubated at 37.5˚C for 18–24 hours. To increase the likelihood of selecting single strain *E. coli* colonies, selected colonies were well isolated and consistent in colour and morphology. The goal was to select 6 colonies per site, although this was not possible in a few sites that had low *E. coli* concentrations. With 6 colonies selected, a strain that composes 25% of the population of *E. coli* in the sample is 80%

likely to be selected at least once. Increasing the number of selected colonies gives diminishing returns in terms of the likelihood of sampling at least one isolate of a strain with a given prevalence (see S3 Fig). Two key factors constrained out total sample size: 1) we were limited to sampling one set per day because of the long distances between sites and the need to walk to many of the homesteads for collecting PoU samples, and 2) our field lab was not equipped for freezing samples nor doing clean DNA extractions, thus limiting the window of time available to us before samples had to be transported to a better-equipped lab to proceed with DNA extraction.

Following incubation, *E. coli* growth on the agar was carefully scraped with a sterile inoculation loop, with care taken to minimise inclusion of agar and off-colour growth (pink growth observed in 12% of samples and colourless growth in 3% of samples). The *E. coli* growth was mixed into 1 mL of DNA Shield (Zymo Research R1100) for preservation in a sterile microcentrifuge tube and stored in a fridge before transport to the Kenya Medical Research Institute (KEMRI) lab in Kilifi.

## DNA extraction and whole genome sequencing

DNA were extracted within 1 to 3 weeks of sampling using a Zymo Research Quick-DNA Mini-prep kit. We adjusted the recommended protocol for monolayer cells to suit *E. coli* that is already lysed by preservation in DNA Shield. Thus, we combined 175 μL of sample lysate with 525 μL of genomic lysis buffer and then proceeded with the monolayer cell protocol as recommended by Zymo Research. Samples were normalised to 5 ng following DNA quantification using a Qubit dsDNA HS Assay Kit and Qubit 3.0 Fluorometer (Life Technologies). We proceeded to library preparation with Illumina Nextera XT DNA Sample Preparation kit as per manufacturer instructions with half reaction alteration.

Following tagmentation and indexing, we did a size selection bead clean-up using 0.6x AMPure XP beads (Beckman Coulter) to select for >500 bp fragments. The libraries were then quantified using Qubit and size distribution was determined on a 2100 Bioanalyser using the High Sensitivity DNA Kit (Agilent Technologies). We proceeded to manual normalization bringing all samples to 2 nm and thereafter pooling the libraries. The pooled libraries were then denatured, spiked with 8% Phix, and run on an Illumina MiSeq platform using the 600 cycles v3 reagent kit with an output of 2 x 200 bp. We did two runs, with 59 libraries pooled in the first run and 68 in the second. One library that was poor quality in the first run, was sequenced again in the second.

## Bioinformatics

There are no known biomarkers that can definitively distinguish whether an *E. coli* isolate comes from a naturalised population or other source [13], so we used multiple characteristics including phylogroup, sequence type, allelic diversity, and presence of virulence and antibiotic resistance genes as suggestive evidence of likely isolate origin. The sequencing reads were processed using the Nullarbor pipeline [36]. Reads were filtered and trimmed using Trimmomatic [37] and only reads that were >100 bp with PHRED quality score >20 were retained. Kraken2 was used for species identification [38] and SPAdes v3.13.1 was used for de novo genome assembly [39].

The assembled genomes were assigned as *E. coli sensu stricto* (phylogroups A, B1, B2, C, D, F, E, or G), *Escherichia* cryptic clades I-V, *E. fergusonii*, or *E. albertii* using the ClermonTyping *in silico* approach based on standard PCR assays and Mash genome distance estimation [40, 41]. The threshold for minimal nucleotides for a contig to be included in the analysis was set at 100. Twelve of the assembled genomes were overlarge (ranging from 5.8 to 11.9 Mbp) so, suspecting chimeric genomes, we conducted Benchmarking Universal Single-Copy Ortholog

(BUSCO) assessment [42, 43] to check the assembled genomes for completeness and duplication. For the non-chimeric genomes, we used Roary [44] for pan-genome analysis. With Roary, genes that are present once in every isolate are combined in a multiple FASTA alignment, enabling phylogenetic tree construction from the core genes. We used FastTree version 2.1.10 SSE3 [45], with generalised time reversible model for nucleotide alignment, to construct an approximately-maximum-likelihood phylogenetic tree. We used GrapeTree [46] to visualise the tree including metadata. For comparison with our sample strains, we included 14 strains from the ClemonTyping Mash database [40] in our analysis, chosen to reflect the diversity of *Escherichia spp*.

Multi-locus sequence typing was done through Nullarbor using the PubMLST database, and virulence and antimicrobial resistance (AMR) gene identification was done using the Abricate package [47] by screening contigs against the VFDB [48] and NCBI AMRFinderPlus [49] databases, respectively. For multi-locus sequence typing, we chose the Achtman MLST scheme [50], which uses genes *adk*, *fumC*, *gyrB*, *icd*, *mdh*, *purA*, and *recA*, because it is most congruent with an established phylogeny for *E. coli* that is based on whole genome sequencing [51]. In addition to the Nullarbor typing, the raw sequencing reads were also run through the MLST screening tool hosted by the Centre for Genomic Epidemiology (CGE) [52] for confirmation of the allele identifications.

## Statistics

To investigate possible relationships between allelic diversity and sample source, we segregated our sampling sites into groups based on location (PoC or PoU) and *E. coli* concentrations (lower or higher) on the day that the isolates were sampled, as shown in Table 2. The cut-off between 'lower' or 'higher' was set at 50 CFU/100mL to balance the number of isolates in each group as evenly as possible. Focussing on the 7 Achtman MLST scheme genes, we used an approach developed for estimating average population heterozygosity from a small number of individuals [53] by calculating the genetic diversity ($H$) of each group based on the diversity of alleles ($h_j$) as:

$$h_j = [1 - \Sigma p_i^2]\left[\frac{n}{n-1}\right] \tag{1}$$

**Table 2. Site groupings based on source type and concentration of *E. coli* on the day that the isolates were sampled.**

| Groups | Sites Included in Each Group | No. of Sets | No. of Samples | No. of MLSTs | No. of Isolates | Median *E. coli*[*] | Min *E. coli*[*] | Max *E. coli*[*] |
|---|---|---|---|---|---|---|---|---|
| All | all sites | 9 | 25 | 46 | 108[+] | 45 | 1 | 2420 |
| PoU | PoU sites | 7 | 15 | 30 | 59 | 16 | 1 | 2000 |
| PoC | PoC sites | 7 | 10 | 24 | 49 | 64 | 2 | 2420 |
| Higher | sites with [>50] | 9 | 11 | 26 | 57 | 460 | 50 | 2420 |
| Lower | sites with [<50] | 5 | 14 | 22 | 51 | 13 | 1 | 45 |
| PoU-H | S1C2U1, S2C1U1, S5C1U3, S7C1U1, S8C1U1, S8C1U2 | 5 | 6 | 15 | 31 | 125 | 50 | 2000 |
| PoU-L | S1C1U2, S2C1U2, S2C2U1, S2C2U2, S2C2U3, S3C1U3, S5C1U2, S6C1U1, S6C1U3 | 5 | 9 | 16 | 28 | 10 | 1 | 17 |
| PoC-H | S3C2, S4C2-A/B, S8C1, S9C1 | 4 | 5 | 14 | 26 | 1120 | 84 | 2420 |
| PoC-L | S1C1, S2C1, S2C2, S2C3, S6C1 | 3 | 5 | 10 | 23 | 42 | 2 | 45 |

[*]Units are MPN/100mL.

[+]This includes 5 chimeric isolates that contained only one match for each MLST gene with perfect identity matches for each allele as explained in the following sequencing overview section.

where $p_i$ is the frequency of allele $i$ at locus $j$ and n is the number of isolates. Genetic diversity (*H*) was then calculated as the average diversity of alleles using:

$$H = \frac{\Sigma h_j}{m} \qquad (2)$$

where *m* is the total number of loci. We assessed the significance of differences in the diversity of the groups using permutation tests derived from the Strasser-Weber framework for conditional inference procedures [54] and performed using the 'coin' package in R version 3.6.1 [55]. We used the general independence test function with asymptotic null distribution as computed by the randomised quasi-Monte Carlo method [56].

## Results and discussion

### Sequencing overview

The sampling effort targeted 6 isolates for each of the 44 sites, but fewer were sequenced for 16 sites and none were sequenced for 20 sites. This was because of low or no concentration of *E. coli* in the sample water (S1C1, S1C1U1/U3-6, S1C2, S1C2U2-3, S2C2U1, S2C3, S3C1, S3C1U1-2/4, S4C1, S4C1U1-2, S5C1, S5C1U1, S6C1, S7C1) or high concentration of thermo-tolerant coliforms (TTCs) preventing clean selection of 6 colonies (S1C2U1, S2C2U3, S3C1U3, S4C2-A, S5C1U2, S5C1U3, S6C1U2, S7C1U2). Other issues preventing sequencing of isolates included TTC contamination of the agar plate (1 isolate each from S3C2 and S7C1U1); limited growth on the agar plate (1 isolate from S2C1U2); inadequate DNA extraction (1 isolate from S4C2-B); and sequencing library preparation failure (1 isolate from S2C1U1).

A total of 125 libraries were successfully sequenced (see S2 Table for the full list), including 4 duplicates. The duplicates were consistent in phylogroup and sequence type. They are not otherwise included in the results. Six of the libraries were contaminated with non-*Escherichia* DNA, five from S1C1U2 contained *Cronobacter sakazakii* and one from S5C1U2 contained *Klebsiella pneumoniae*. These libraries were removed from further analysis. The remaining 115 libraries had 628,219 reads per library on average (SD = 145 199), with mean read length of 177 bp (SD = 7), mean PHRED quality score of 31 (SD = 0.7), and mean depth of 23 (SD = 6). BUSCO assessment of the genomes assembled from these 115 libraries identified 12 chimeric genomes. Excluding the chimeric genomes, the reads from each library assembled into an average of 246 contigs (SD = 141), with mean genome size of 4.8 Mbp (SD = 0.2).

Four of the chimeric genomes have multiple perfect allele matches for at least one of the Achtman MLST genes, confirming them as chimeras of multiple MLSTs. A further three of them have imperfect identity matches for multiple alleles (ranging from 86.5% to 99.8% identity match) and do not correspond to known sequence types. These 7 chimeras with multiple MLSTs or imperfect allele matches were excluded from further analysis. The remaining 5 chimeric genomes contain only one match for each MLST gene with perfect identity matches for each allele. These single MLST chimeric genomes are included in the ClermonTyping and MLST results (total of 108 isolates) but were excluded from the pangenome and virulence and AMR analyses (total of 103 isolates).

### Pan-genome and phylogeny

The pan-genome of our isolates and the 14 references strains that we included for comparison has a total of 25,526 genes, including 1794 core genes (in $\geq$ 99% of the genomes), 1201 soft core genes (in $\geq$ 95%), 2152 shell genes (in $\geq$ 15%), and 20,379 cloud genes (in < 15%). The phylogenetic tree generated from the core genes (Fig 2) reflects the ClermonTyping

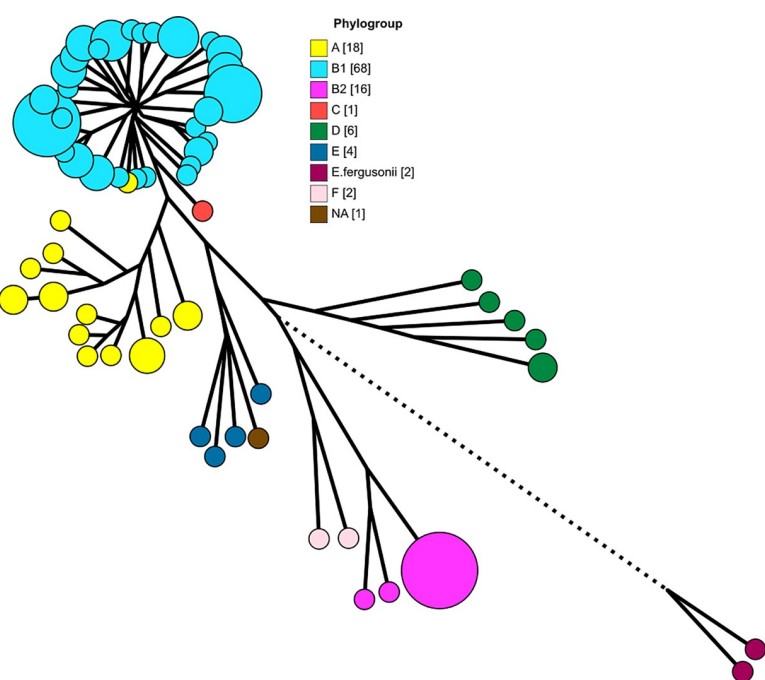

**Fig 2. Phylogenetic tree with nodes coloured by phylogroup as determined by ClermonTyping.** The bracketed numbers in the image keys indicate the number of isolates in each category. The tree is scaled to 125% to make visual parsing easier. Branches with length less than 0.0006 nucleotide substitutions per site were collapsed and the size of the nodes is scaled to indicate the number of isolates each one encompasses. The branch length for *E. fergusonii* was shortened from 0.05 to 0.03 substitutions per site.

characterisation of our isolates except for isolate 2-8B (MLST 3519), which the *in silico* assays classified as phylogroup A but was closer to B1 isolates in the Mash estimation and the phylogenetic tree. We note that the MLST 3519 entries in the Warwick Enterobase database are also classified as phylogroup A with AxB1 lineage [22]. When the phylogenetic tree nodes are colour-coded by set (Fig 3), we observe that the evolutionary similarity of the isolates is not related to the water system that they came from.

The ClermonTyping analysis classified 69 isolates from 21 sites as belonging to phylogenetic group B1, making B1 the most represented group in the sample set. The second-most prevalent is group A with 15 isolates from 8 sites, followed by group B2 with 14 isolates from 3 sites, group D with 6 isolates from 4 sites, and group E with 3 isolates from 3 sites. One isolate from S3C2 could not be classified by ClermonTyping, but the phylogenetic tree indicates that it belongs in phylogroup E (Fig 2). We did not identify cryptic clade *E. coli*, which are the group most strongly associated with environmental origins [13, 34].

The association between phylogroup and strain origin varies based on diet, hygiene, animal domestication status, and morphological and socioeconomic factors [8]. Some localised studies have found differences in the relative frequency of phylogroups by strain origin [57], but others have not [58]. A review of results from both higher and lower income countries in Europe, Africa, the Americas, Asia, and Australia [8] found that phylogroup B1 strains were dominant in animals (41%), followed by A (22%), B2 (21%), and D strains (16%); whereas A strains were most common in humans (40.5%), followed by B2 (25.5%) and then B1 and D strains (17% each). Phylogroup B1 strains, which may be more common in animal faeces than in humans, comprised 78% of our PoC isolates and 50% of our PoU isolates. In contrast, A strains, which may be more common in humans, comprised 4% of our PoC isolates and 22% of our PoU isolates.

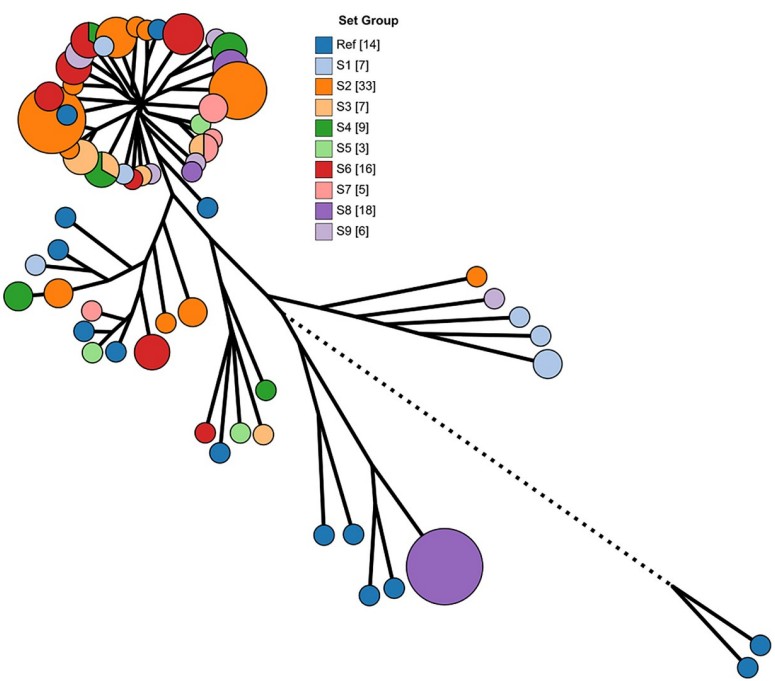

**Set Group**
- ■ Ref [14]
- ■ S1 [7]
- ■ S2 [33]
- ■ S3 [7]
- ■ S4 [9]
- ■ S5 [3]
- ■ S6 [16]
- ■ S7 [5]
- ■ S8 [18]
- ■ S9 [6]

**Fig 3. Phylogenetic tree with nodes coloured by sample set.**

The dominance of phylogroup B1 and A strains in our samples does not necessarily indicate recent faecal contamination. Strains from B1 and A are better generalists and are more prevalent in freshwater samples than other strains [59]. B1 strains, especially, have been found to survive best in the environment [7, 60], and in freshwater specifically [9, 21, 61, 62], and phenotypes linked to environmental survival are relatively prevalent in B1 isolates [63, 64]. In contrast, B2 and D isolates do not survive well and are under-represented in freshwater [21, 59]. Less is known about phylogroup E, which is a small set of formerly unassigned strains that are relatively uncommon, historically difficult to cluster phylogenetically, and generally understudied [50, 65] (except for the O157:H7 serotype, which is clinically important but is excluded by typical culture methods that rely on β-glucuronidase activity).

## Virulence genes

For the 103 non-chimeric isolates, a total of 184 virulence genes were identified with 100% identity and coverage. The complete list per isolate can be found in S2 Table. These genes represent 8 functional groups including secretion (67 genes), adherence (54 genes), iron uptake (46 genes), chemotaxis and motility (6 genes), invasion (5 genes), immune evasion (3 genes), autotransport (2 genes), and toxin production (1 gene). Most of the genes occur with low frequency across the isolates, and the number of genes possessed by any one isolate ranges from 34 to 94, with most having between 40 and 60 (S4 Fig). Isolates from phylogenetic group A have the least virulence genes (mean = 43, SD = 6.7), and isolates from groups B2 (63, 0.5), E (67, 1.7), and D (75, 9.2) have higher numbers than most B1 isolates (55, 10.1).

Due to the plasticity of the *E. coli* genome, it can be challenging to define and identify pathovars, but some combinations of virulence genes have been linked to different *E. coli* pathotypes [48, 66]. Genes for Shiga toxin production were not identified, ruling out presence of enterohemorrhagic *E. coli* (EHEC) or Shiga toxin-producing *E. coli* (STEC). Eight isolates from phylogroup B1 have multiple enteropathogenic *E. coli* (EPEC) associated virulence genes

for adherence, autotransport, invasion, iron uptake, motility, secretion, and toxin production, but they lack key bundle-forming pili (*bfp*) and intimin (*eae*) genes, suggesting that they are neither typical nor atypical EPEC [67]. Similarly, all twenty isolates from phylogroups D or B2 have multiple genes that are associated with uropathogenic *E. coli* (UPEC) and / or neonatal meningitis-associated *E. coli* (NMEC), but all are missing key genes such as *fdeC* for adherence or *cnf1* for toxin production.

None of our isolates were identified as complete pathovars, but the presence of virulence genes is informative, nonetheless, since virulence genes are more prevalent in strains isolated from humans and the trend is conserved within phylogroups [8, 21]. Four phylogroup B1 isolates have exceptionally high numbers of virulence genes (>80), suggesting that they are human derived. All four were isolated from PoU sites (S3C1U3, S6C1U1, S6C1U3). Furthermore, the phylogroup B2 and D isolates possessed more virulence genes than most of the A and B1 isolates. Taken together with evidence of their relatively poor survival in the environment [21, 59], this further supports that the PoC and PoU sites that had B2 or D isolates were subject to recent contamination. This applies to three PoC sites that were poorly protected from contamination (S1C1, an open reservoir; S8C1, an open dug well; S9C1, an unsealed concrete tank) and 4 household sites (S1C2U1, S2C1U2, S8C1U1, S8C1U2).

## Antimicrobial resistance genes

For the 103 non-chimeric isolates, a total of 24 AMR genes were identified with 100% identity and coverage, with every isolate having at least one AMR gene associated with resistance to aminoglycosides (*aph(6)-Id*, *aph(3'')-Ib*, *aadA1*, *aadA5*, *aac(3)-IId*), beta-lactams (*blaCTX-M-14*, *blaEC-5/8/13/15/18*, *blaTEM-1*), erythromycin (*mph(A)*), quaternary ammonium compounds (*qacEdelta1*), sulphonamides (*sul1/2*), tetracycline (*tet(A)*, *tet(B)*), or trimethoprim (*dfrA1*, *dfrA5*, *dfrA14*, *dfrA17*). Additionally, arsenite resistance gene *arsB-mob*, which codes for an arsenite efflux pump, was found in 82 isolates. Arsenic resistance can develop in response to use of arsenicals in antimicrobial drugs or in response to naturally occurring arsenic in the environment [68]. Concentrations of arsenic were low at the PoC sites, ranging from 0.04 to 0.95 ppb (measurement by inductively coupled plasma mass spectrometry; unpublished data from the monthly monitoring programme). As such, geogenic arsenic is unlikely to explain the prevalence of the *arsB-mob* gene. Furthermore, the absence of additional arsenic resistance genes suggests that the presence of arsB-mob may not be related to drug use or geogenic arsenic, rather research into broad arsenic resistance in prokaryotes points to ancestral gene clusters as a likely explanation [69].

Excluding *arsB-mob*, at least one AMR gene was found in each of the 103 isolates. Most of the genes occur with low frequency across the isolates (see S2 Table for a list of AMR genes per isolate), with 88 isolates having only 1 gene. AMR genes are transferrable between bacteria via plasmids and environmental reservoirs of resistance genes are widely recognised, including freshwater and drinking water systems [70, 71]. Nevertheless, multiple AMR genes may be more common in *E. coli* strains sourced from human as opposed to animal or naturalised populations [13, 72, 73]. Four PoU sites had isolates with multiple AMR genes including 3 phylogroup D isolates with 7 to 12 genes each (S1C2U1) and 6 phylogroup B1 isolates with 4 to 7 genes each (S2C2U1, S3C1U3, S7C1U1). Additionally, two PoC sites (S4C2 and S9C1, both unsealed concrete tanks) had phylogroup B1 isolates with 4 to 7 AMR genes each.

## Multi-locus sequence types

Using the Achtman MLST scheme, a total of 40 previously known sequence types were identified. As detailed in S3 Table, the number of entries for these MLSTs in the Warwick

Enterobase database (as of May 2020) ranged from 1 to 7763; 10 of the MLSTs had not previously been identified in Kenya; and 6 of the isolates have MLST allele combinations that were not represented in the database. These new combinations are listed in S2 Table, we have labelled them New_1 to New_6.

We refer to a system and its associated PoCs and PoUs as a set. The main discriminating factor between the sequence types is the set that the isolate came from (Fig 4). The sequence types generally do not overlap between sets, except in the cases of ST10, SW180, ST345, and ST216. We are also interested in the overlap in sequence types between matched PoC and PoU sites. There was no overlap for set 1, but sets 2, 6, and 8 did have overlap. For sets 3 and 5, no *E. coli* was isolated from the PoCs (except S3C2, which is an unsealed concrete water tank at a school that is unrepresentative of the wider distribution system). For sets 4, 7, and 9, *E. coli* was only isolated from one site each.

Logically, five scenarios may dictate the population of *E. coli* in PoU water, these are permutations of three factors: presence or absence of *E. coli* in the PoC water, whether PoC strains retain culturability in PoU water, and whether post-collection hygiene conditions introduce new strains. Although the isolates analysed in our study allow only a partial view of the diversity of *E. coli* strains at each sampling site, comparison of the MLSTs isolated from matched PoC and PoU sites suggests that multiple PoUs exemplified each of the five scenarios (Fig 5).

In scenarios 3, 4, and 5 no additional health hazard is introduced at the household level: *E. coli* in PoU water is determined by PoC water quality and persistence or abatement of strains in storage containers. Abatement of *E. coli* does not equate to abatement of pathogens (decreased health risk), so PoC results should be prioritised in these scenarios. In scenarios 1 and 2, inadequate hygiene at the household level does influence PoU *E. coli* population, either as the exclusive driver (scenario 2) or in combination with persistence of PoC *E. coli* strains (scenario 1). Only five of the households that we sampled had overlap in MLSTs between PoC and PoU (scenarios 1 and 3), this points to strain persistence but is not evidence of regrowth. Nevertheless, the results are interesting in light of research that a) found increase in *E. coli* between PoC and PoU was unrelated to household-level sanitary or hygiene factors [31] and b] demonstrated regrowth in household storage containers within 48 hours [75]. Conversely, Scenario 2, post-collection contamination, appears to be the most common for the households that we sampled, which is consistent with studies that relate PoU water quality deterioration to unsafe storage [28, 76].

The health implications of post-collection contamination are debated. In households like those in our study area, with low levels of access to sanitation and hygiene facilities, *E. coli* and other faecal indicators and pathogens are widespread on surfaces and in food produce [77] and it is likely that strains circulate between humans, animals, and the domestic environment [8, 78]. Large-scale randomised control trials investigating the impact of water, sanitation, and hygiene interventions on health outcomes for communities in Kenya, Bangladesh and elsewhere have demonstrated the importance of considering multiple faecal exposure pathways [79]. In a household setting, where there are multiple pathways for exposure to pathogens that are circulating in the household environment, focussing on only one of these pathways (stored water) is unlikely to reduce the burden of enteric disease in the household [80, 81]. In contrast, a contaminated water distribution system may pose a unique threat as a pathway for spreading disease between households and, in some cases, between communities.

## Allelic diversity

In addition to directly comparing MLSTs, we queried the differences between sites further by using the MLST genes to analyse differences in the diversity of the isolates from the lower

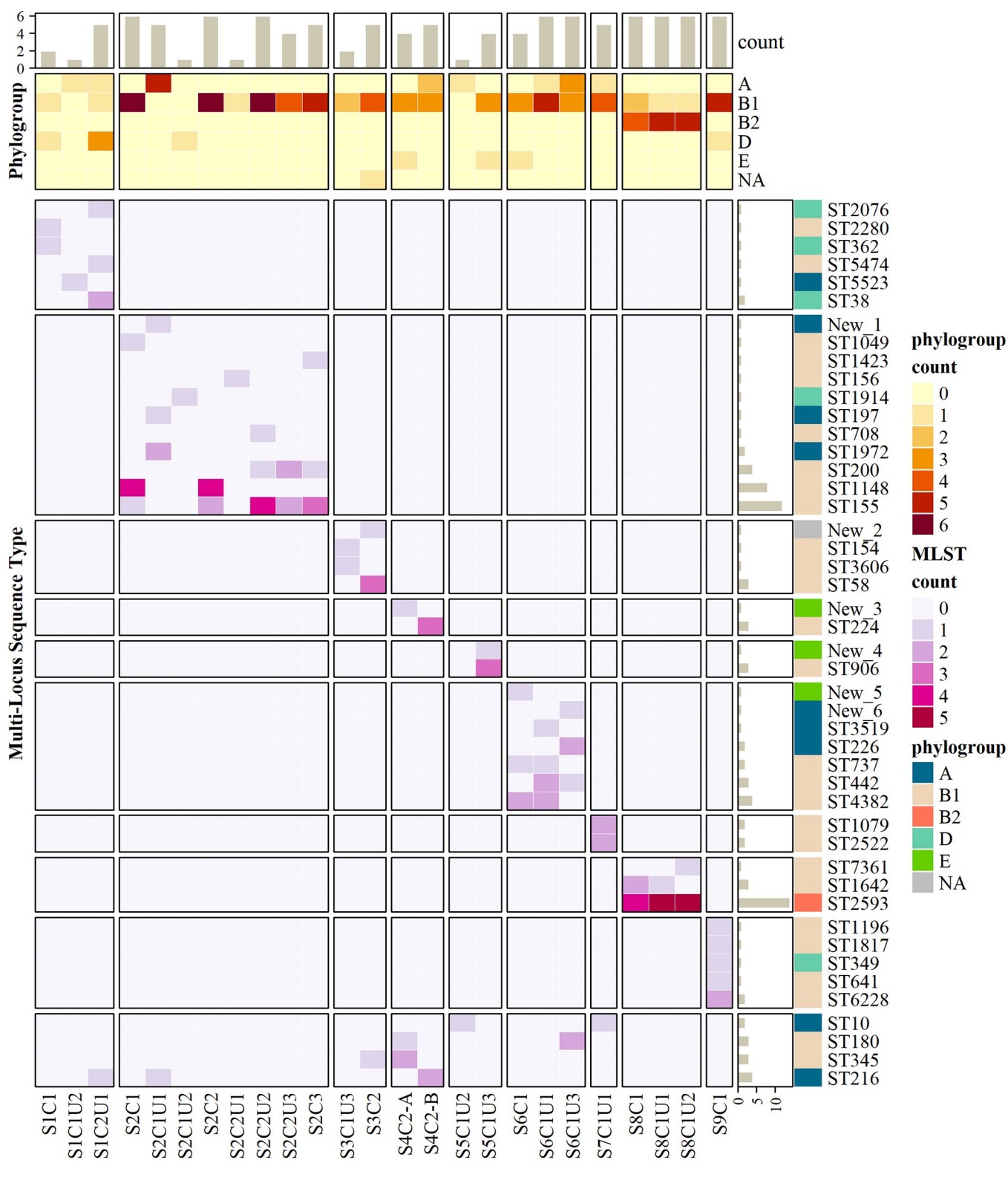

**Fig 4. Heatmap of phylogenetic groups and MLSTs identified for each sampling site.** Colour gradients indicate number of isolates. Bar chart annotations show total counts of isolates per MLST (right) and per site (top). The heatmap was created with R package 'ComplexHeatmap' [74].

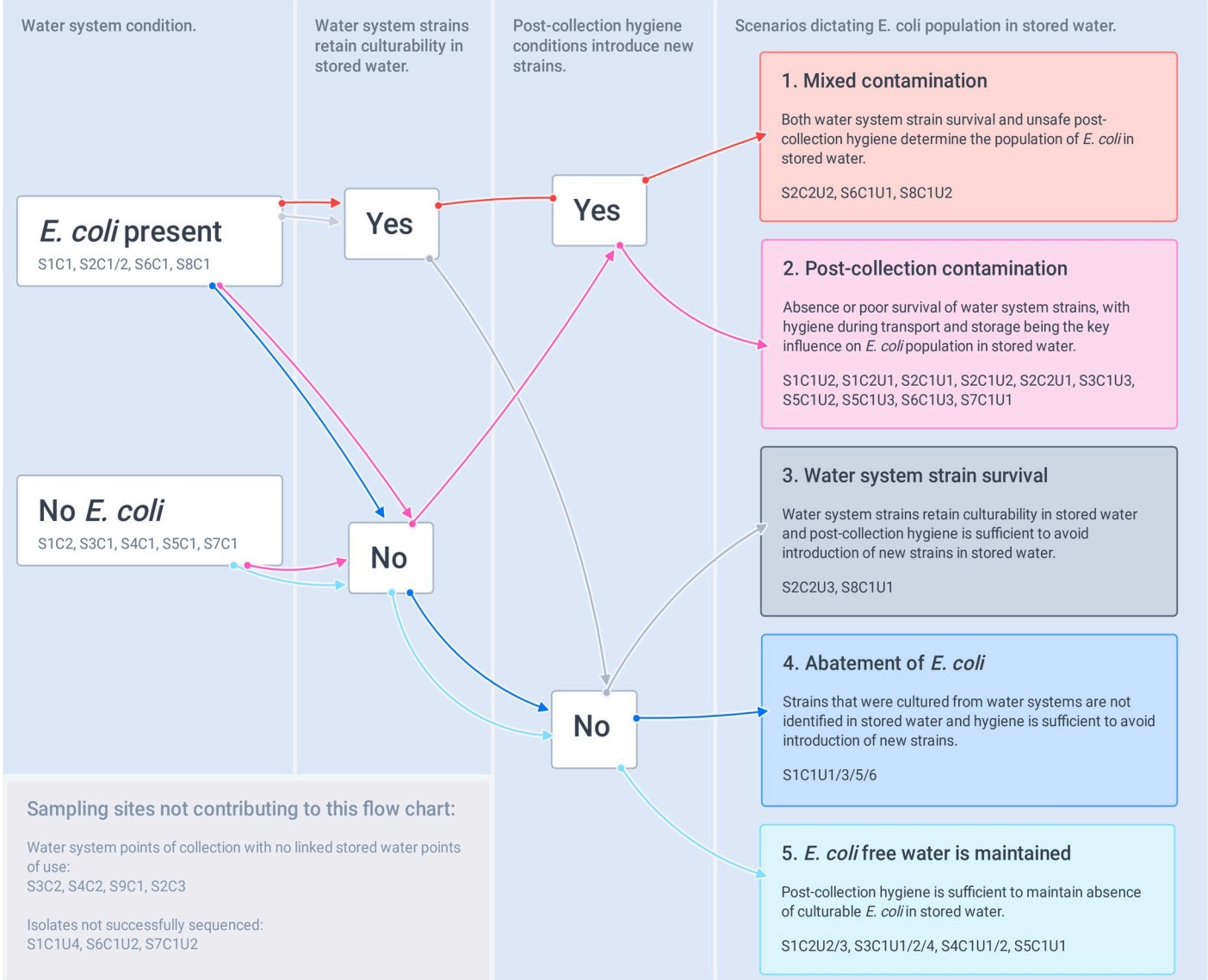

**Fig 5. Five scenarios theorised to explain the population of *E. coli* in stored water at point of use.** Scenarios are based on sustained presence of strains that were isolated from the water supply system and addition of new strains during post-collection transport and storage.

(<50 CFU/100mL) and higher (>50 CFU/100mL) concentration PoC and PoU sites. We found all loci are polymorphic in all groupings (Table 3) and the diversity of alleles ($h_j$) ranges from 0.32 to 0.91. A permutation test comparing all four of the sub-groups (PoU-H, PoU-L, PoC-H, and PoC-L) found that the grouping has an effect on diversity (z = 2.47, p = 0.0499); more specifically, pairwise tests found that the PoC-L group is less diverse than both the PoC-H group (z = 2.12, p = 0.03) and the PoU-H group (z = 2.29, p = 0.02). No other pairwise differences are significant, including comparisons of the PoU sites versus the PoC sites (z = 1.13, p = 0.26), the high concentration sites versus the low concentration sites (z = 1.82, p = 0.07), or the PoU-L sites versus the PoU-H sites (z = 1.4, p = 0.17).

That the allelic diversity of *E. coli* isolates from low concentration PoCs is significantly less than from high concentration sites is interesting in light of studies that found lower allelic

**Table 3. Genetic diversity of alleles (*h<sub>j</sub>*) and average genetic diversity (*H*) for groupings based on source type (PoC vs PoU) and *E. coli* concentration level (higher or lower than 50 MPN/100mL).**

| Groups | H | Diversity of Alleles ($h_j$) | | | | | | |
|---|---|---|---|---|---|---|---|---|
| | | *adk* | *fumC* | *gyrB* | *icd* | *mdh* | *purA* | *recA* |
| All | 0.80 | 0.63 | 0.83 | 0.90 | 0.87 | 0.85 | 0.70 | 0.81 |
| PoU | 0.82 | 0.72 | 0.88 | 0.88 | 0.89 | 0.86 | 0.70 | 0.82 |
| PoC | 0.76 | 0.51 | 0.76 | 0.90 | 0.84 | 0.81 | 0.71 | 0.76 |
| Higher | 0.82 | 0.74 | 0.84 | 0.89 | 0.87 | 0.84 | 0.77 | 0.80 |
| Lower | 0.70 | 0.44 | 0.78 | 0.84 | 0.82 | 0.80 | 0.54 | 0.64 |
| PoU-H | 0.81 | 0.79 | 0.84 | 0.85 | 0.85 | 0.82 | 0.80 | 0.74 |
| PoU-L | 0.72 | 0.52 | 0.85 | 0.81 | 0.87 | 0.78 | 0.44 | 0.76 |
| PoC-H | 0.81 | 0.65 | 0.80 | 0.91 | 0.89 | 0.86 | 0.73 | 0.81 |
| PoC-L | 0.62 | 0.32 | 0.67 | 0.76 | 0.77 | 0.75 | 0.62 | 0.45 |

diversity and greater genome similarity in samples from environmental sources compared to samples from faecal sources [13, 82, 83]. The comparison suggests that PoC sites with low concentration of *E. coli* may be more associated with naturalised *E. coli*. Although not significantly different from the high concentration sites, the low concentration PoU sites also had lower allelic diversity and, therefore, may have been less affected by recent contamination. This interpretation of allelic diversity supports use of *E. coli* risk categories [26] and aligns with research findings that indicate a threshold effect, with significant increase in diarrhoeal disease burden only associated with high concentrations (>1000/100 mL) of *E. coli* [84].

Interpretations of *E. coli* results, however, should not rely exclusively on concentration: the health implications of differences in *E. coli* concentrations are context dependent and naturalisation is not the only process that confuses the relationship between *E. coli* and health hazard. For example, the water for set 1 is sourced from an open reservoir system, which has animal and human activity in the catchment area and does not include treatment. Thus, the low concentration of *E. coli* in S1C1, and absence in S1C2, may be better interpreted as indicating poor survival of *E. coli* in the reservoir–likely due to a combination of predation, competition, UV radiation, and absence of surfaces for biofilm formation [9, 10]–rather than absence of faecal contamination. Furthermore, one of the isolates from S1C1 was from phylogroup D with numerous virulence genes, a likely indicator of recent human faecal contamination. Additionally, for sets 3 and 5 the water is chlorinated in the distribution system, which gives more assurance of safety, but some pathogens are more resistant to chlorine than *E. coli* [26]. Generally, concentrations of *E. coli* do not correlate with concentrations of pathogens: the transport and survival patterns of *E. coli* vary considerably from those of faecal pathogens, particularly viruses and protozoa which tend to be more robust [25]–so the likelihood that water has been contaminated with faecal matter must be prioritised over *E. coli* sampling results.

## Summary and recommendations

Although definitive attribution is not possible, the strains that most likely originated from human and/or recent faeces were found in poorly protected PoC water (4 sites including an unfenced open reservoir, unfenced open dug well, and two unsealed concrete tanks) or PoU water (12 out of 30 PoU sites). These were the 34 isolates from phylogroups A, B2, and D, and the 16 from phylogroup B1 with >80 virulence genes or multiple AMR genes. The other B1 isolates with fewer virulence genes account for almost half of our sample (48%), likely because B1 strains are generally better adapted to the freshwater environment. Allelic diversity comparisons suggest that naturalised *E. coli* may be particularly relevant at PoC sites with lower *E.*

*coli* concentrations (<50 / 100mL). And for PoU sites, analysis based on five theorised PoU *E. coli* population scenarios underscores the difficulty of interpreting health risk from grab samples.

Placing our findings in relation to the literature, we develop two main recommendations. Firstly, we emphasise the inadequacy of judging hazard based on single *E. coli* samples at either PoC or PoU. Tracking sanitary conditions and *E. coli* concentrations over time can inform a more reliable understanding of hazard. In addition to *E. coli* sampling, rapid, *in situ* measurements such as turbidity or tryptophan-like fluorescence may be useful for high frequency tracking of water quality variability; although they have their own limitations, these measures can indicate process changes in water systems including, for example, response to rainfall events [85, 86], which may help differentiate between naturalised *E. coli* and contamination events. And regardless of water quality measures, sanitary inspection is needed to confirm the current and prospective safety of a system. Studies have found weak or no correlations between sanitary inspection scores and microbial water quality as measured by faecal indicator bacteria (FIB) [87], but this does not diminish the importance of the inspections given what we know of FIB results having multiple possible explanations.

Secondly, we recommend that PoC and PoU *E. coli* samples should not be compared directly in terms of their health hazard implications. *E. coli* in PoC water (especially when concentration is high) should be prioritised for interventions with a focus on water safety management to provide safe water at the PoC. On the other hand, PoU samples are more difficult to interpret because uncertainty is introduced by variability in: PoC quality, persistence of strains, and post-collection hygiene. Positive *E. coli* samples at household level could indicate no additional health hazard but, conservatively, they should be interpreted as indicating a hazardous household environment, generally. Effective intervention at the household-level requires a multi-pathway approach that goes beyond water treatment and safe storage.

## Limitations and further research

The growth of *E. coli* is influenced by physicochemical characteristics of water such as nutrient levels, salinity, and temperature, as well as microbiome characteristics such as competition and predation [9, 10]. But given the sample size of our study, we are not able to query the impact of these factors on the balance between growth and die-off of *E. coli* strains. Similarly, our study did not focus on temporal change in *E. coli* populations. Only one site was sampled twice: S4C2 had no overlap in MLSTs between the two samples taken two weeks apart. This suggests that continual contamination is driving the population dynamics at this site rather than persistent dominance of strains in biofilms or otherwise. The site is a poorly protected concrete tank with multiple openings situated in a market square. Furthermore, the water at the site is saline (median 4.1 mS/cm), and salinity is known to inhibit *E. coli* survival in water [88]. Thus, the conditions at this site seem to enable ongoing input of new *E. coli* whilst discouraging *E. coli* survival and growth, which could explain the lack of overlap in the time-separated samples. A larger study incorporating a temporal dimension would improve insight into *E. coli* population dynamics in water systems over time–including the impact of sanitary conditions and physiochemical and microbiome characteristics. Additionally, a larger study would enable better characterisation of strain diversity within samples if more isolates per sample were analysed (S3 Fig).

Another avenue for further work is prompted by the prevalence of phylogroup B1 isolates in our samples, given their association with animal faeces [8]. Multiple studies have now emphasised the importance of animal management as a key sanitary factor influencing drinking water safety [89–91]; however, the importance of zoonotic transmission is not well

established in the water, sanitation and hygiene (WASH) literature and recent models relating health outcomes to WASH factors have excluded zoonotic pathways in part due "to data scarcity on animal faeces and animal presence" [92 p279]. Further work in this space would be valuable.

Finally, the limitations of genomic characterisation for informing on strain origin is a key constraint of this study–we are able to comment on the likelihood of isolates being naturalised or recently sourced from faeces but cannot definitively identify them as such. To-date there have been few studies focussing on genomic characterisation of *E. coli* from drinking water supplies, but as the collective dataset grows it will enable metanalyses and more robust statistics that will improve our ability to distinguish naturalised strains and better understand the origins, diversity, and dynamics of *E. coli* populations in water supplies.

## Supporting information

**S1 Fig. Box-and-whisker plot of monitoring programme pH results for selected PoCs.** The boxes show median values and span lower to upper quartiles, the whiskers show the lowest and highest datums within 1.5 times the interquartile range.
(TIFF)

**S2 Fig. Box-and-whisker plot of monitoring programme conductivity results for selected PoCs.** The boxes show median values and span lower to upper quartiles, the whiskers show the lowest and highest datums within 1.5 times the interquartile range.
(TIFF)

**S3 Fig. Likelihood of strain selection given number of selected colonies and strain prevalence.**
(TIFF)

**S4 Fig. Stacked bar chart displaying the number of virulence genes per isolate by phylogroup.**
(TIFF)

**S1 Table. List of water sets.** Includes points of collection (PoC) and points of use (PoU) with median *E. coli*, pH and conductivity (EC) from monitoring programme results.
(XLSX)

**S2 Table. List of sequenced isolates.** Includes Sequencing, Assembly, Phylogroup, MLST, Virulome, and Resistome Results.
(XLSX)

**S3 Table. List of identified MLSTs grouped by water system.**
(XLSX)

## Acknowledgments

The authors thank Cliff Nyaga, Musenya Sammy, Mbogo Mwaniki and the rest of the Rural Focus Limited and REACH Kenya teams for their support in making the fieldwork possible.

## Author Contributions

**Conceptualization:** Saskia Nowicki, Katrina J. Charles.

**Data curation:** Saskia Nowicki, Etienne P. de Villiers.

**Formal analysis:** Saskia Nowicki, Etienne P. de Villiers, George Githinji.

**Funding acquisition:** Saskia Nowicki, Katrina J. Charles.

**Investigation:** Saskia Nowicki.

**Methodology:** Saskia Nowicki, Zaydah R. deLaurent, Etienne P. de Villiers, George Githinji.

**Project administration:** Saskia Nowicki.

**Resources:** Saskia Nowicki, Zaydah R. deLaurent.

**Software:** Etienne P. de Villiers.

**Supervision:** Katrina J. Charles.

**Validation:** Saskia Nowicki, Etienne P. de Villiers.

**Visualization:** Saskia Nowicki.

**Writing – original draft:** Saskia Nowicki.

**Writing – review & editing:** Saskia Nowicki, Zaydah R. deLaurent, Etienne P. de Villiers, George Githinji, Katrina J. Charles.

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
