## [Decision Letter · Decision Letter 0]

25 Nov 2020

PONE-D-20-30642

The Utility of Escherichia coli as a Contamination Indicator for Rural Drinking Water: Evidence from Whole Genome Sequencing

PLOS ONE

Dear Dr. Nowicki,

Thank you for submitting your manuscript to PLOS ONE. After careful consideration, we feel that it has merit but does not fully meet PLOS ONE’s publication criteria as it currently stands. Therefore, we invite you to submit a revised version of the manuscript that addresses the points raised during the review process.

We look forward to receiving your revised manuscript.

Kind regards,

Andrew C Singer, Ph.D.

Academic Editor

PLOS ONE

Journal Requirements:

Additional Editor Comments (if provided):

I thank you for a very well received manuscript. The reviewers are agreed in the value of your study and have made some useful suggestions for how the presentation and interpretation can be improved (particularly Reviewer 2). I recommend you take on board these suggestions--but in the case where you do not agree you offer a full rebuttal. I look forward to receiving your revised manuscript.

Reviewers' comments:

Reviewer's Responses to Questions

**Comments to the Author**

1. Is the manuscript technically sound, and do the data support the conclusions?

Reviewer #1: Yes

Reviewer #2: Yes

Reviewer #3: Yes

2. Has the statistical analysis been performed appropriately and rigorously? 

Reviewer #1: Yes

Reviewer #2: Yes

Reviewer #3: Yes

3. Have the authors made all data underlying the findings in their manuscript fully available?

Reviewer #1: Yes

Reviewer #2: No

Reviewer #3: Yes

4. Is the manuscript presented in an intelligible fashion and written in standard English?

Reviewer #1: Yes

Reviewer #2: Yes

Reviewer #3: Yes

5. Review Comments to the Author

Reviewer #1: The Utility of Escherichia coli as a Contamination Indicator for Rural Drinking Water: Evidence from Whole Genome Sequencing

The authors used whole genome sequencing to access the suitability of E. coli as an indicator of recent faecal contamination. Though they are no definitive biomarkers for microbial source tracking of E. coli, they used MLST, allelic diversity, virulence and antibiotics genes to group the isolates and predict their likely sources. Interestingly was the presents of isolates likely to have originated from naturalized populations, which is important considering E. coli is universally used as an indicator of recent faecal contamination. The overall findings support the use of E. coli and encourages monitoring that accounts for sanitary conditions and temporal variability.

General Comment

The manuscript is well written, and the data is presented well. However, they are some minor issues which might be addressed.

Below find some of the points

On author list you can add a period on Katrina J. Charles

Are the periods necessary after subtopics, e.g. line 99? Kindly check, if it’s not adjust throughout the manuscript

Line 210: you can add the concentration

Line 229: I don’t think the bracket is necessary

Line 230: SPades should be SPAdes; also reference for SPAdes and Kraken2 is necessary

Line 251: I think Acthman MLST should be Achtman MLST, you should also correct on Line 415

Line 318: you can slightly expand the discrepancy core genome phylogeny and Typing of 2-8B

Line 455: you can add the Pickering et al., 2012 reference; which is (70), I guess

Line 561: WASH should be in full at the first time of mentioning

It will also be nice if you could link the isolates to their SRA identifiers

Reviewer #2: Overall

The work demonstrates the use of whole genome sequencing of E. coli isolates to identify strain type, phylogroup, virulence factors, and antimicrobial resistance as a potential method for improving understanding of water quality issues in Kenya. The work is innovative in its attempt to use genomics to improve understanding of water contamination in the household environment. Although the work appears generally rigorous, the interpretations of some of the findings (most notably source attribution) should be reconsidered as discussed in the major comments. It is also unclear where the data (besides the sequences) are, can the authors please specify a repository?

Major Comments:

1) The authors attribute strains to sources based on previous review work on the relative proportions of phylogroups within different sources. Phylogroups alone, or in conjunction with virulence factors, can not be used to definitively attribute sources (yet) due to high levels of variation in strain diversity. Additionally, these reviews may be geographically biased based on underrepresentation of studies in LMICs.

A number of studies looking at E. coli diversity in low income settings have been conducted showing the inadequacy of phylogroups, and the authors could refer to this literature (see detailed comments). The authors should consider removing references to strain sources given the weak evidence, or provide stronger evidence.

2) Methods should be described in more detail, or references provided with the methodological details. See detailed comments.

3) The authors pose an interesting conceptual model for the causes and consequences of E. coli strain diversity in water at the point of collection vs. at the point of use. However, I think the authors overemphasize the potential role of E. coli persistence and/or growth in water and biofilms. The authors should consider their work in relation to prior work on this (see Levy at all in detailed comments).

Detailed Comments:

Line 2 – preferred or dominant?

Line 7 – suggest n = 14, n = 30.

Line 16 – 18 – I do not understand this sentence, can the authors simplify or explain in more detail.

Line 53-60 – suggest removing the sentences about novelty to kenya, novelty as defined by absence of other research is not sufficient to justify research importance and is generally less important to PloS One. Further, research on naturalized E. coli populations in Kenya was arguably done by Olilo et al. https://doi.org/10.1007/s40974-018-0081-3, though the question of identifying naturalized vs. fecal isolates remains open. Indeed the work presented here may be entirely fecal populations.

Lines 135 – please include description of methods/equipment used to measure pH, conductivity. How do you know the sites with high conductivity are only used when better alternatives are not available?

Line 141 – please report methods for E. coli detection and quantification, including positive and negative control frequency and results.

Line 143 – I prefer log10 transformation to ln transformation because it is more widely used and more intuitive to understand. Please consider switching units to log10. Alternative, consider reporting the x-axis for figure 1 with the true values, with logn-scaling.

Line 156 – how were users surveyed to find out information about the cleaning regime? Please report methods.

Line 183 – suggest referring to E. coli as thermotolerant E. coli, if they are incubated at elevated temperatures, as 44.5C is not standard recommendations for m-ColiBlue24, and inclusion of this data in metaanalyses in the future may be biased due to potentially decreased sensitivity for detection of E. coli.

Line 191 – this is well done, that you calculated and determined the number of isolates that should be tested to be broadly representative of the isolates in the sample.

Table 2 – Mean of untransformed data is often biased for E. coli concentrations (which tend to be log-distributed). Suggest including a column for Median or log10 transformed concentrations, as well as Range.

Line 297 – please clarify, were the samples removed from futher analysis or were the sequences cleaned to remove non- e. coli DNA?

Line 307 – unidentified strain types are common amongst environmental isolates from countries that are underrepresented in sequence databases. See Montealegre et al. 2019 DOI: 10.1128/mSphere.00704-19. The paper is conceptually similar and may be of interest to the authors.

Line 341: Although it is good the authors not the dominance of strains by strain types from the review, I do not think there is enough evidence to suggest that “animal mfeces may be an important source in both PoC and PoU”. I think the authors should state the association without trying to infer causality. For example, “our strains were predominately B1 phylogroup, which is dominant in animals … “. Additionally, it would be good to note the percent of human strains that are B1. Of note, the review may be heavily skewed geographically, as there tends to be underrepresentation of LMICs in sequencing databases. In aforementioned Montealegre et al. work, isolates from humans, cattle, chickens, and soil were all mapped to B1 and A, suggesting these phylogroups are not sufficiently descriptive of source.

Line 366 – its unclear what the point of Figure 4 is. Consider removing, moving to SI, or making it more clear why this is included.

Line 367 – very clear description of the dataset as it relates to pathovars.

Line 379 – “since virulence genes”… again, I would consider geographic bias in reviews of sequencing databases

Line 385 – I am unsure that the review referenced (55) is sufficiently up to date, in other work by Montealegre et al. DOI: 10.1128/AEM.01978-18, E. coli isolate growth/persistence is similar amongst B1 and A isolates. Again, there may be geographic bias in studies.

Line 403 – is arsenic resistance classified as antimicrobial resistance?

Line 426 – heatmap created with R package not necessary reference for this figure, stating that R was used in analyses is sufficient.

Figure 6 – this is a very fascinating approach, but I am unsure E. coli strain typing can be or should be used in this way without a better description or understanding of strain type diversity. What is the likelihood that strain types are circulating in the household are similar to the strain types in the PoC?

I also find this chart a bit difficult to follow. Are the colored lines distinct, or can they be crossed? For example, it looks like the grey line must be followed (culturability in stored water but no introduction post-collection of new strains suggests scenario 2) water system survival). In contrast, the pink line has no path to No for Strain culturability. Should there be a pink line from E. coli Present to No?

It might be interesting to make the width of the lines an indicator of the proportion of samples within each scenario.

Line 428 – the term sets is difficult to follow here, maybe it was introduced earlier, suggest re-introducting what a set is.

Line 470 – this is clearly borderline significant, I suggest stating that.

Line 471 – to what extent is the difference in diversity driven by an inability to culture 6 isolates from POC-L group? Are the groups balanced in the number of isolates from each sample?

Line 508 – I do not yet see the evidence to support this. Phylogroup can not (and should not) be used to identify source. Julian et al. (DOI: 10.1128/AEM.03214-14) attempted to attribute E. coli strains to sources using phylogroup and virulence traits, and were largely unsuccessful.

Gomi et al. DOI: 10.1021/es501944c developed a library-dependent method for source tracking E. coli based on WGS, and were able to identify specific markers for sources, but this approach may be specific to the location studies. I suggest the authors reconsider this discussion, and limit the attribution of strains to sources without further data on strains circulating amongst humans and animals in the area (similar to the aforementioned Gomi and Montealegre references).

Line 519 – in the samples the authors studied, how did strain-specific analysis compare to the findings based only on concentrations? What fraction of samples would be classified as low household-contamination risk using strain typing that were classified as high risk based on counts?

Line 534 – Levy et al. DOI 10.1289/ehp.11296 is a seminal paper that attributes overwhelmingly the contamination post collection to the household environment, showing growth in the containers is not a meaningful contributor to contamination. The authors should consider acknowledging this.

A limitation of the study is that strain diversity in the samples is limited to 6 isolates, and so can not be fully characterized and strain types can not be attributed to human and animal isolates.

Reviewer #3: The manuscript is very well written and is virtually error free, the authors must be commended for this. The research was performed using sound approaches and methods, looked at the issue in a comprehensive manner, and the rationale used during discussion and interpretation was sound. The findings in this manuscript are also very insightful and I am sure that the water sector would find the manuscript to be of value.

There is one sentence in the abstract that requires revision, as I could not understand this sentence without reading the manuscript itself. I had another water scientist read the abstract and they also could not make sense of that sentence. Given how good the rest of the manuscript is it would be a pity if this were not revised. The sentence in the abstract reads "Based on strain presence-absence comparisons, five scenarios dictated E. coli population in water at the point of use, underscoring the difficulty of interpreting sampling results from these sites." Without the context given in the manuscript itself this sentence does not make sense, and just confuses the reader as he/she is introduced to the research in the abstract. My suggestion is to heavily revise or remove that sentence. That is my only request for minor revision, for the rest this manuscript is of such high quality that I whole heartedly recommend it for publication.

6. PLOS authors have the option to publish the peer review history of their article (what does this mean?). If published, this will include your full peer review and any attached files.

Reviewer #1: No

Reviewer #2: No

Reviewer #3: **Yes: **Wouter le Roux (Senior Researcher, Water Centre, CSIR, South Africa)

---

## [Author Response · Author response to Decision Letter 0]

21 Dec 2020

Dear editor Singer and reviewers, 

Thank you for the opportunity to revise and resubmit our article. We have revised the manuscript to address your feedback and suggestions. We feel it is stronger now and are grateful for your input and attention to detail. Especially considering the extra difficulties this year has involved, we thank you for the time you have taken to engage with our work. Below we explain how we addressed each of your points. Please note that the page and line numbers that we refer to correspond with the marked-up version of the revised manuscript. 

Responses to Reviewer 1:

On author list you can add a period on Katrina J. Charles

Thank you for catching this. We have added the period.

Are the periods necessary after subtopics, e.g. line 99? Kindly check, if it’s not adjust throughout the manuscript 

 Thank you for highlighting this. We have removed the periods.

Line 210: you can add the concentration 

Thank you for the suggestion. The samples were normalised to 5 ng before library prep. We have now added this in the text (line 234).

Line 229: I don’t think the bracket is necessary 

Thank you for catching this. We removed the bracket.

Line 230: SPades should be SPAdes; also reference for SPAdes and Kraken2 is necessary 

Thank you. We have corrected the typo and added the references.

Line 251: I think Acthman MLST should be Achtman MLST, you should also correct on Line 415 

Thank you. We have corrected the spelling.

Line 318: you can slightly expand the discrepancy core genome phylogeny and Typing of 2-8B 

Thank you for highlighting this, we have expanded on it in the text (line 343). 

Line 455: you can add the Pickering et al., 2012 reference; which is (70), I guess 

Thank you for catching this, we corrected the citation format error.

Line 561: WASH should be in full at the first time of mentioning 

Thank you, we have made the correction.

It will also be nice if you could link the isolates to their SRA identifiers 

Good point, thank you! We uploaded the read files to the European Nucleotide Archive. We have now added the ENA identifiers to the list of isolates in Supplementary Table 2. 

Responses to Reviewer 2:

Comments from the reviewer questionnaire:

The authors have not made all data underlying the findings in their manuscript fully available. It is also unclear where the data (besides the sequences) are, can the authors please specify a repository? 

The monitoring data that were used to select sampling sites for this study were generated through a monitoring programme operated by a rural water maintenance service provider with government oversight. We have altered the text to clarify this (line 111). We have permission to use summary information in publications, but do not have permission to publicise the raw data.

Major concerns:

1) general comment: Although the work appears generally rigorous, the interpretations of some of the findings (most notably source attribution) should be reconsidered as discussed in the major comments. The authors attribute strains to sources based on previous review work on the relative proportions of phylogroups within different sources. Phylogroups alone, or in conjunction with virulence factors, can not be used to definitively attribute sources (yet) due to high levels of variation in strain diversity. Additionally, these reviews may be geographically biased based on underrepresentation of studies in LMICs. 

A number of studies looking at E. coli diversity in low income settings have been conducted showing the inadequacy of phylogroups, and the authors could refer to this literature (see detailed comments). The authors should consider removing references to strain sources given the weak evidence, or provide stronger evidence.

 general response: Thank you, we appreciate and share your concern that the discussion of our results not be interpreted as definitive source tracking, but rather that in considering our results in the context of the wider literature we are discussing provisory indications not certainties. We noted in the abstract and in the main text that definitive source tracking is not possible, but your feedback helpfully made clear to us that this point needed to be emphasised more strongly and that some of our statements warranted revision. We have detailed the changes that we made in the responses below.

Detailed comments related to major concern 1:

Line 341: Although it is good the authors not the dominance of strains by strain types from the review, I do not think there is enough evidence to suggest that “animal mfeces may be an important source in both PoC and PoU”. I think the authors should state the association without trying to infer causality. For example, “our strains were predominately B1 phylogroup, which is dominant in animals … “. Additionally, it would be good to note the percent of human strains that are B1. Of note, the review may be heavily skewed geographically, as there tends to be underrepresentation of LMICs in sequencing databases. In aforementioned Montealegre et al. work, isolates from humans, cattle, chickens, and soil were all mapped to B1 and A, suggesting these phylogroups are not sufficiently descriptive of source. 

 Thank you for this detailed comment and suggestions. We have made revisions including both rephrasing the statements as you recommend and adding the percent of human strains that are B1 in the review study (line 373). We agree that LMICs are generally underrepresented in databases. Though it is notable that the review we reference, Tenaillon et al. 2010, drew from studies that were geographically diverse including 6 continents and lower income countries such as Benin, Mali, Gabon and others. Nevertheless, in keeping with Tenaillon et al. 2010 we have stated that phylogroup distribution varies based on diet, hygiene, animal domestication status, and morphological and socioeconomic factors (line 366). For better balance, we have now used Julian et al. 2015 as an example of studies that have shown no source driven difference in phylogenetic distribution.

Line 508 – I do not yet see the evidence to support this. Phylogroup can not (and should not) be used to identify source. Julian et al. (DOI: 10.1128/AEM.03214-14) attempted to attribute E. coli strains to sources using phylogroup and virulence traits, and were largely unsuccessful. Gomi et al. DOI: 10.1021/es501944c developed a library-dependent method for source tracking E. coli based on WGS, and were able to identify specific markers for sources, but this approach may be specific to the location studies. I suggest the authors reconsider this discussion, and limit the attribution of strains to sources without further data on strains circulating amongst humans and animals in the area (similar to the aforementioned Gomi and Montealegre references). 

 We have revised the summary section (now starting line 567) in keeping with the changes we made in the Pan-genome and phylogeny section as noted above.

Line 379 – “since virulence genes”… again, I would consider geographic bias in reviews of sequencing databases

 Thank you for highlighting this consideration. We have chosen two references for this point (which is now at line 416), the first is Tenaillon et al. 2010 which we chose because they reviewed studies from 6 continents including lower and higher income countries. The second is Touchon et al. 2020, which we chose because, although the isolates were all from Australia, the study has a large sample size and was done recently so it references other useful studies (such as those noting that B1 isolates from water have low virulence factor counts) if a reader wants to do a deep dive on this point. 

Line 385 – I am unsure that the review referenced (55) is sufficiently up to date, in other work by Montealegre et al. DOI: 10.1128/AEM.01978-18, E. coli isolate growth/persistence is similar amongst B1 and A isolates. Again, there may be geographic bias in studies.

 Thank you for raising this question. We have now referenced Touchon et al. 2020 here as well to provide a more current reference that continues to find B2 and D isolates are less successful in environmental conditions (line 421). This follows on from a paragraph in the preceding ‘Pan-genome and phylogeny’ section (line 380) that discusses a series of studies from varying locations/contexts that found B1 and A survive better in the environment and B1 in particular does better in freshwater. The findings of Montealegre et al. show no B2 or D isolates from soil samples, which is in accordance with the point we are making here.

2) general comment: Methods should be described in more detail, or references provided with the methodological details. See detailed comments.

Lines 135 – please include description of methods/equipment used to measure pH, conductivity. How do you know the sites with high conductivity are only used when better alternatives are not available?

Line 141 – please report methods for E. coli detection and quantification, including positive and negative control frequency and results.

 This information for pH, conductivity, and E. coli monitoring has now been added to the methods section (line 147). 

Line 156 – how were users surveyed to find out information about the cleaning regime? Please report methods.

 This information has now been added (line 171). 

Line 297 – please clarify, were the samples removed from futher analysis or were the sequences cleaned to remove non- e. coli DNA?

 Thanks for highlighting this. We have now specified that the libraries were removed from further analysis (line 322).

3) general comment: The authors pose an interesting conceptual model for the causes and consequences of E. coli strain diversity in water at the point of collection vs. at the point of use. However, I think the authors overemphasize the potential role of E. coli persistence and/or growth in water and biofilms. The authors should consider their work in relation to prior work on this (see Levy at all in detailed comments).

 general response: Thank you for this helpful feedback. As detailed in the response column below, we have worked to address this concern by revising the presentation and explanation of Figure 5 (formerly Figure 6), more clearly acknowledging limitations in the text, and better contextualising our results with reference to the literature including Levy et al. 2008.

Detailed comments related to major concern 3:

Figure 6 – this is a very fascinating approach, but I am unsure E. coli strain typing can be or should be used in this way without a better description or understanding of strain type diversity. What is the likelihood that strain types are circulating in the household are similar to the strain types in the PoC? I also find this chart a bit difficult to follow. Are the colored lines distinct, or can they be crossed? For example, it looks like the grey line must be followed (culturability in stored water but no introduction post-collection of new strains suggests scenario 2) water system survival). In contrast, the pink line has no path to No for Strain culturability. Should there be a pink line from E. coli Present to No? It might be interesting to make the width of the lines an indicator of the proportion of samples within each scenario.

 Thank you for this feedback. We agree that this analysis has important limitations, but we think it contributes usefully to the discussion on interpreting E. coli grab samples from household water. We have revised the introduction to Figure 6 (now Figure 5) to more clearly acknowledge that the results are limited and suggestive as opposed to conclusive (line 474). We have redrawn the coloured lines in the figure and rearranged the order of the scenarios so that there is a distinct path to each scenario, and it is more intuitive to follow. We prefer not to weight the lines because this is intended more as a conceptual rather than quantitative analysis, though we have listed the sites that align with each scenario on the figure and noted in the text that post-collection contamination was the most common scenario (line 496). 

Line 534 – Levy et al. DOI 10.1289/ehp.11296 is a seminal paper that attributes overwhelmingly the contamination post collection to the household environment, showing growth in the containers is not a meaningful contributor to contamination. The authors should consider acknowledging this.

 Thank you for highlighting this. Although Levy et al. 2008 conclude that recontamination is more important than regrowth in the households that they sampled, they do not rule out the possibility that regrowth was also occurring, and they acknowledge that regrowth has been demonstrated in other contexts. We have included more discussion on the role of E. coli persistence or growth in HH water and have referenced Levy at al. 2008 (line 497). We also deleted former line 534 in the summary section to avoid overemphasising the role of regrowth and thereby detracting from the larger point. 

Other detailed comments from reviewer 2:

Line 2 – preferred or dominant? 

In many places it is both dominant and preferred, but we chose ‘preferred’ because in some places thermotolerant or total coliforms are still used in lieu of E. coli specifically. 

Line 7 – suggest n = 14, n = 30. 

Thank you for the suggestion, we have made this change.

Line 16 – 18 – I do not understand this sentence, can the authors simplify or explain in more detail. 

Thank you for this feedback. We replaced the sentence with a more straightforward and broader description of the key discussion theme.

Line 53-60 – suggest removing the sentences about novelty to kenya, novelty as defined by absence of other research is not sufficient to justify research importance and is generally less important to PloS One. Further, research on naturalized E. coli populations in Kenya was arguably done by Olilo et al. https://doi.org/10.1007/s40974-018-0081-3, though the question of identifying naturalized vs. fecal isolates remains open. Indeed the work presented here may be entirely fecal populations. 

Thanks for bringing our attention to this point. Our intention here was to highlight that a meta-analysis of non-clinical E. coli genetic diversity (like the Australian example) is not currently possible for Kenya, or Africa more broadly, due to a lack of primary studies. This was to provide a view on the current literature and pre-empt queries on why we did not try to address our research questions with a larger n, meta-analytic approach. We have made revisions to clarify this point (line 61). We considered referencing Olilo et al. since their study was conducted in Kenya, but we decided against it because their focus on use of manure in fields is quite divergent from our interest in drinking water supplies and they didn’t generate whole genome sequences. 

Line 143 – I prefer log10 transformation to ln transformation because it is more widely used and more intuitive to understand. Please consider switching units to log10. Alternative, consider reporting the x-axis for figure 1 with the true values, with logn-scaling. 

Thanks for this feedback, we have changed it to log10.

Line 183 – suggest referring to E. coli as thermotolerant E. coli, if they are incubated at elevated temperatures, as 44.5C is not standard recommendations for m-ColiBlue24, and inclusion of this data in metaanalyses in the future may be biased due to potentially decreased sensitivity for detection of E. coli.

Thank you for this suggestion. We considered whether incubation at 44.5C would introduce systematic bias in the study, in particular we did not want to disadvantage naturalised E. coli. We noted this in the text (line 202). From the literature, we were satisfied that E. coli are generally thermotolerant (making ‘thermotolerant E. coli’ redundant), with maximal growth rate and optimal growth temperature (41-42C) being consistent and unrelated to phylogenetic affiliation, including for the cryptic clade phylogroups, and tolerance extending into the upper forties. Incubation at 35C with m-ColiBlue allows a simultaneous test for total coliforms (red colonies) and E. coli (blue colonies), but we were not interested in TCs.

Line 191 – this is well done, that you calculated and determined the number of isolates that should be tested to be broadly representative of the isolates in the sample. 

Thank you.

Table 2 – Mean of untransformed data is often biased for E. coli concentrations (which tend to be log-distributed). Suggest including a column for Median or log10 transformed concentrations, as well as Range. 

Good point! We have replaced the mean and standard error of the mean with median and range.

Line 307 – unidentified strain types are common amongst environmental isolates from countries that are underrepresented in sequence databases. See Montealegre et al. 2019 DOI: 10.1128/mSphere.00704-19. The paper is conceptually similar and may be of interest to the authors. 

Yes, we agree that it is not unusual to find unknown MLSTs in environmental samples from countries that are underrepresented in databases. We don’t feel that a reference is warranted here but thank you for prompting us to consider the Montealegre et al. 2020 paper. We have referenced it later in the manuscript to improve our discussion of E. coli at household level (line 511).

Line 366 – its unclear what the point of Figure 4 is. Consider removing, moving to SI, or making it more clear why this is included. 

Thank you for this feedback. We have moved the figure into supplementary information and included mean and standard deviation values in the text instead.

Line 367 – very clear description of the dataset as it relates to pathovars. 

Thank you.

Line 403 – is arsenic resistance classified as antimicrobial resistance? 

 Thank you for highlighting the lack of clarity here. The arsB gene is discussed in this section because it is included in the NCBI AMRFinderPlus database that we screened our isolates against. Its inclusion in the database is likely due to historical and current use of arsenicals in antimicrobials such as salvarsan (arsphenamine), melarsoprol, and arsinothricin. But arsB has also been linked to ancestral gene clusters and given the absence of any other arsenic resistance genes in our isolates, it is likely that the presence of arsB in our samples is not related to resistance to antimicrobial drugs or geogenic arsenic. This is why we excluded it from further analysis/discussion. We have added two sentences and a reference to help clarify this (lines 433 and 438).

Line 426 – heatmap created with R package not necessary reference for this figure, stating that R was used in analyses is sufficient. 

Thank you for this recommendation. We would like to retain the citation because in the documentation for the ComplexHeatmap package, the creator has specifically asked that the citation be included if the package is used to create outputs for publications.

Line 428 – the term sets is difficult to follow here, maybe it was introduced earlier, suggest re-introducting what a set is. 

Thank you for this feedback. The term was introduced at line 133. We have now repeated the explanation at line 460 for ease of reading.

Line 470 – this is clearly borderline significant, I suggest stating that. 

Thank you for the suggestion. We have removed ‘significant’ and allow the p value to speak for itself (line 526).

Line 471 – to what extent is the difference in diversity driven by an inability to culture 6 isolates from POC-L group? Are the groups balanced in the number of isolates from each sample?

Thanks for this query. Lower E. coli concentration was not the only reason that 6 isolates were not always cultured for each sample, higher concentration samples had issues with TTC growth for example (more details in the manuscript, line 308). The median and mean number of isolates from each sample are: PoU-H: median 5, mean 5.2; PoU-L: median 4, mean 3.1; PoC-H: median 5, mean 5.2; PoC-L: median 5, mean 4.6. PoU-L had higher diversity than PoC-L, so the number of isolates per sample doesn’t correspond to the diversity. We’ve added the No. of Samples to Table 2 so that it can be compared with No. of Isolates if others have this query.

Line 519 – in the samples the authors studied, how did strain-specific analysis compare to the findings based only on concentrations? What fraction of samples would be classified as low household-contamination risk using strain typing that were classified as high risk based on counts?

Thank you, these are interesting questions. In this paper we have not developed a new risk classification scheme based on strain typing. Our results support use of the WHO risk categories and prioritising higher risk water (line 543), especially with sanitary inspection and regular sampling to better capture context and improve interpretation (line 547, line 583). We note that interpretation of E. coli results at household level is additionally difficult because uncertainty is introduced by variability in: PoC quality, persistence of strains, and post-collection hygiene. PoC samples are clearer indicators of water supply safety, whereas positive E. coli samples from household water can conservatively be interpreted as indicating a hazardous household environment generally (line 603). 

A limitation of the study is that strain diversity in the samples is limited to 6 isolates, and so can not be fully characterized and strain types can not be attributed to human and animal isolates.

We agree that these are important limitations. In the limitations section, we have now reiterated that within sample diversity would be better characterised if more samples per isolate were analysed (line 622). The attribution issue is addressed as a key limitation in the final paragraph starting at line 632.

Responses to Reviewer 3:

There is one sentence in the abstract that requires revision, as I could not understand this sentence without reading the manuscript itself. I had another water scientist read the abstract and they also could not make sense of that sentence. Given how good the rest of the manuscript is it would be a pity if this were not revised. The sentence in the abstract reads "Based on strain presence-absence comparisons, five scenarios dictated E. coli population in water at the point of use, underscoring the difficulty of interpreting sampling results from these sites." Without the context given in the manuscript itself this sentence does not make sense, and just confuses the reader as he/she is introduced to the research in the abstract. My suggestion is to heavily revise or remove that sentence. 

Thank you for this helpful feedback. We replaced the sentence with a more straightforward and broader description of the key discussion theme.

---

## [Editor Report · Decision Letter 1]

11 Jan 2021

The utility of Escherichia coli as a contamination indicator for rural drinking water: Evidence from whole genome sequencing

PONE-D-20-30642R1

Dear Dr. Nowicki,

We’re pleased to inform you that your manuscript has been judged scientifically suitable for publication and will be formally accepted for publication once it meets all outstanding technical requirements.

Kind regards,

Andrew C Singer, Ph.D.

Academic Editor

PLOS ONE
---

## [Editor Report · Acceptance letter]

13 Jan 2021

PONE-D-20-30642R1 

The utility of *Escherichia coli* as a contamination indicator for rural drinking water: Evidence from whole genome sequencing 

Dear Dr. Nowicki:

I'm pleased to inform you that your manuscript has been deemed suitable for publication in PLOS ONE. Congratulations! Your manuscript is now with our production department. 

Kind regards, 

on behalf of

Dr. Andrew C Singer 

Academic Editor

PLOS ONE